# Spontaneous skyrmionic lattice from anisotropic symmetric exchange in a Ni-halide monolayer

Danila Amoroso ⬤ [1✉], Paolo Barone ⬤ [1] & Silvia Picozzi[1]

Topological spin structures, such as magnetic skyrmions, hold great promises for data storage applications, thanks to their inherent stability. In most cases, skyrmions are stabilized by magnetic fields in non-centrosymmetric systems displaying the chiral Dzyaloshinskii-Moriya exchange interaction, while spontaneous skyrmion lattices have been reported in centrosymmetric itinerant magnets with long-range interactions. Here, a spontaneous anti-biskyrmion lattice with unique topology and chirality is predicted in the monolayer of a semiconducting and centrosymmetric metal halide, $NiI_2$. Our first-principles and Monte Carlo simulations reveal that the anisotropies of the short-range symmetric exchange, when combined with magnetic frustration, can lead to an emergent chiral interaction that is responsible for the predicted topological spin structures. The proposed mechanism finds a prototypical manifestation in two-dimensional magnets, thus broadening the class of materials that can host spontaneous skyrmionic states.

[1] Consiglio Nazionale delle Ricerche CNR-SPIN, c/o Università degli Studi 'G. D'Annunzio', 66100 Chieti, Italy. ✉email: danila.amoroso@spin.cnr.it

Magnetic skyrmions are localized topological spin structures characterized by spins wrapping a unit sphere, and carrying an integer topological charge $Q$[1,2]. Their topological properties ensure the inherent stability that makes them technologically appealing for future memory devices[3]. Noncoplanarity in the direction of the spin magnetic moments at different lattice sites is a necessary—albeit not sufficient—ingredient to obtain a net scalar spin chirality $\mathbf{s}_i \cdot (\mathbf{s}_j \times \mathbf{s}_k)$, in turn related to the topological invariants that characterize most of the appealing properties of skyrmions[1,2].

While the interplay of competing magnetic interactions may often lead to noncoplanarity, localized skyrmion-like magnetic textures with fixed chirality are generally believed to arise from the Dzyaloshinskii–Moriya (DM) interaction, driven by spin–orbit coupling (SOC) in systems lacking space-inversion symmetry. Such short-range antisymmetric exchange interaction, in fact, acts as a chiral interaction and fixes one specific rotational sense of spins, thus imposing a well-defined chirality to noncollinear and noncoplanar spin textures[4–10].

Conversely, in geometrically frustrated centrosymmetric lattices (such as triangular or Kagome), possible skyrmion-lattice states triggered by competing exchange interaction manifest with various topologies[11–15], as there is no mechanism determining a priori their topology and chirality[16–20]. These states, in fact, generally arise from nonchiral interactions, such as easy-axis magnetic anisotropy, long-range dipole–dipole and/or Ruderman–Kittel–Kasuya–Yosida (RKKY) interactions, and thermal or quantum fluctuations[11–15,21]. Furthermore, skyrmionic spin structures are usually stabilized by external fields in both noncentrosymmetric and centrosymmetric materials. A spontaneous skyrmion-lattice state has been proposed so far only in itinerant magnets displaying amplitude variations of the magnetization[4], where its microscopic origin was attributed to long-range effective four-spin and higher-spin interactions that arise from conduction electrons[6,22–25].

In this work, we show that the spontaneous formation of thermodynamically-stable skyrmionic lattice with a unique, well-defined topology and chirality of the spin texture can also be driven by the anisotropic part of the short-range symmetric exchange, in absence of DM and Zeeman interactions, but assisted only by the exchange frustration. In particular, by performing density functional theory (DFT) and Monte-Carlo (MC) simulations, we report a spontaneous high-$Q$ antiskyrmion lattice and a field-induced topological transition to a standard skyrmion lattice in a potential semiconducting 2D-magnet, the centrosymmetric $NiI_2$ monolayer.

## Results

**Skyrmionic lattice in $NiI_2$.** $NiI_2$ is a centrosymmetric magnetic semiconductor long known for its exotic helimagnetism[26–29]. It belongs to the family of transition-metal-based van der Waals materials recently object of intense research activity due to their intriguing low dimensional magnetic properties[30–39]. A single layer of $NiI_2$ is characterized by a triangular net of magnetic cations at distance $a_0$ and competing ferro (FM)-magnetic and antiferro(AFM)-magnetic interactions, resulting in strong magnetic frustration (Fig. S2 in Supplementary). Our DFT and MC calculations reveal that $NiI_2$ monolayer displays a spontaneous transition below $T_c \simeq 30$ K to a triple-$\mathbf{q}$ state (with $\mathbf{q}_1 = (\delta, \delta)$, $\mathbf{q}_2 = (\delta, -2\delta)$, $\mathbf{q}_3 = (-2\delta, \delta)$ and $\delta \simeq 0.125$, as detailed in Fig. 1). This state consists in a triangular lattice of antibiskyrmions (A2Sk) characterized by a topological charge $|Q| = 2$ with associated vorticity $m = -2$, as shown in Fig. 1; see refs. [18,40,41] for further insights about skyrmionic spin structures. The magnetic unit cell (m.u.c.) of such A2Sk lattice, with lateral size $L_{m.u.c.} \simeq 8a_0$,

comprises three nanoscale antibiskyrmions, each with an approximate diameter of about 1.6 nm (Fig. S6) and surrounded by six vortices with vanishing net magnetization; the central spins ($\mathbf{s}_i$) of the A2Sk have opposite $s_z$ component with respect to the vortices center (in the specific case reported in Fig. 1a, spins in the antibiskyrmion core and in the vortices centers point upward and downward, respectively). When a perpendicular magnetic field ($B_z$) is applied, a conventional Bloch-type skyrmion lattice with $m = 1$ (shown in Fig. 1b) is induced for a finite range of applied fields, before a ferromagnetic state is finally stabilized for large fields. Upon the field-induced transition, all spins surrounding one every two downward vortices align to the field, while the in-plane spin pattern remains substantially unchanged. A sharp topological phase transition thus occurs as the charge $|Q|$ changes from 2 to 1 at the critical ratio $B_z/J^{1iso} > 0.4$ ($J^{1iso}$ being the nearest-neighbor isotropic exchange interaction defined in the next section), while it changes to 0 for $B_z/J^{1iso} \simeq 2.2$, as displayed in Fig. 1d–f. The magnetization $\mathbf{M}$ also exhibits evidences of these phase transitions, signaled by abrupt changes in correspondence of the critical fields; the magnetization saturation, corresponding to all spin aligned with the magnetic field, takes place for $B_z/J^{1iso} > 3.9$, as shown in Fig. 1e.

A similar sequences of topological phase transitions and the spontaneous onset of the A2Sk lattice has been previously predicted in frustrated itinerant magnets described by a Kondo-lattice model on a triangular lattice[24]. Nevertheless, the effective interaction between localized spins mediated by conduction electrons, that has been identified as the driving mechanism for the stabilization of the topological spin textures in such itinerant magnets[25,42], cannot be invoked to explain the A2Sk lattice in semiconducting $NiI_2$. Similarly, the DM interaction has to be excluded, being forbidden by the inversion symmetry of the lattice. As discussed below, here the A2Sk state, and related field-induced state, directly arise from the anisotropic properties of the short-range symmetric exchange. As such, the underlying mechanism is not restricted to itinerant magnets and metallic systems, but rather has a more general validity, as it can apply also to centrosymmetric magnetic semiconductors.

**Underlying microscopic mechanisms.** Magnetic interactions between localized spins $\mathbf{s}_i$ can be generally modeled by the classical spin Hamiltonian

$$H = \frac{1}{2}\sum_{i \neq j} \mathbf{s}_i \mathbf{J}_{ij} \mathbf{s}_j + \mathbf{s}_i \mathbf{A}_i \mathbf{s}_i, \qquad (1)$$

where $\mathbf{A}_i$ and $\mathbf{J}_{ij}$ denote the on-site or single-ion anisotropy (SIA) and the exchange coupling interaction tensors, respectively[43]. The latter is generally decomposed into three contributions[44–46]: the isotropic coupling term $J_{ij}^{iso} = \frac{1}{3}\mathrm{Tr}\mathbf{J}_{ij}$, defining the scalar Heisenberg model $H^{iso} = \frac{1}{2}\sum_{i \neq j} J_{ij}\mathbf{s}_i \cdot \mathbf{s}_j$; the antisymmetric term $\mathbf{J}_{ij}^A = \frac{1}{2}(\mathbf{J}_{ij} - \mathbf{J}_{ij}^T)$, which corresponds to the DM interaction and vanishes in the presence of an inversion center on the spin–spin bond (as realized in the systems under investigation); the anisotropic symmetric term $\mathbf{J}_{ij}^S = \frac{1}{2}(\mathbf{J}_{ij} + \mathbf{J}_{ij}^T) - J_{ij}^{iso}\mathbf{I}$, also referred to as two-site anisotropy ($\mathbf{J}^{two-site\ aniso}$), which is of particular interest here. The operator Tr and the superscript T label the trace and the transpose of the full $\mathbf{J}_{ij}$ matrix; $\mathbf{I}$ is the unit matrix. As the DM interaction, the two-site anisotropy arises from the spin–orbit coupling; while the former favors the canting of spin pairs, the latter tends to orient the spins along given orientations in space. When the principal (anisotropy) axes between different spin pairs are not parallel, an additional frustration in the relative orientation of adjacent spins may lead to noncoplanar magnetic configurations, in

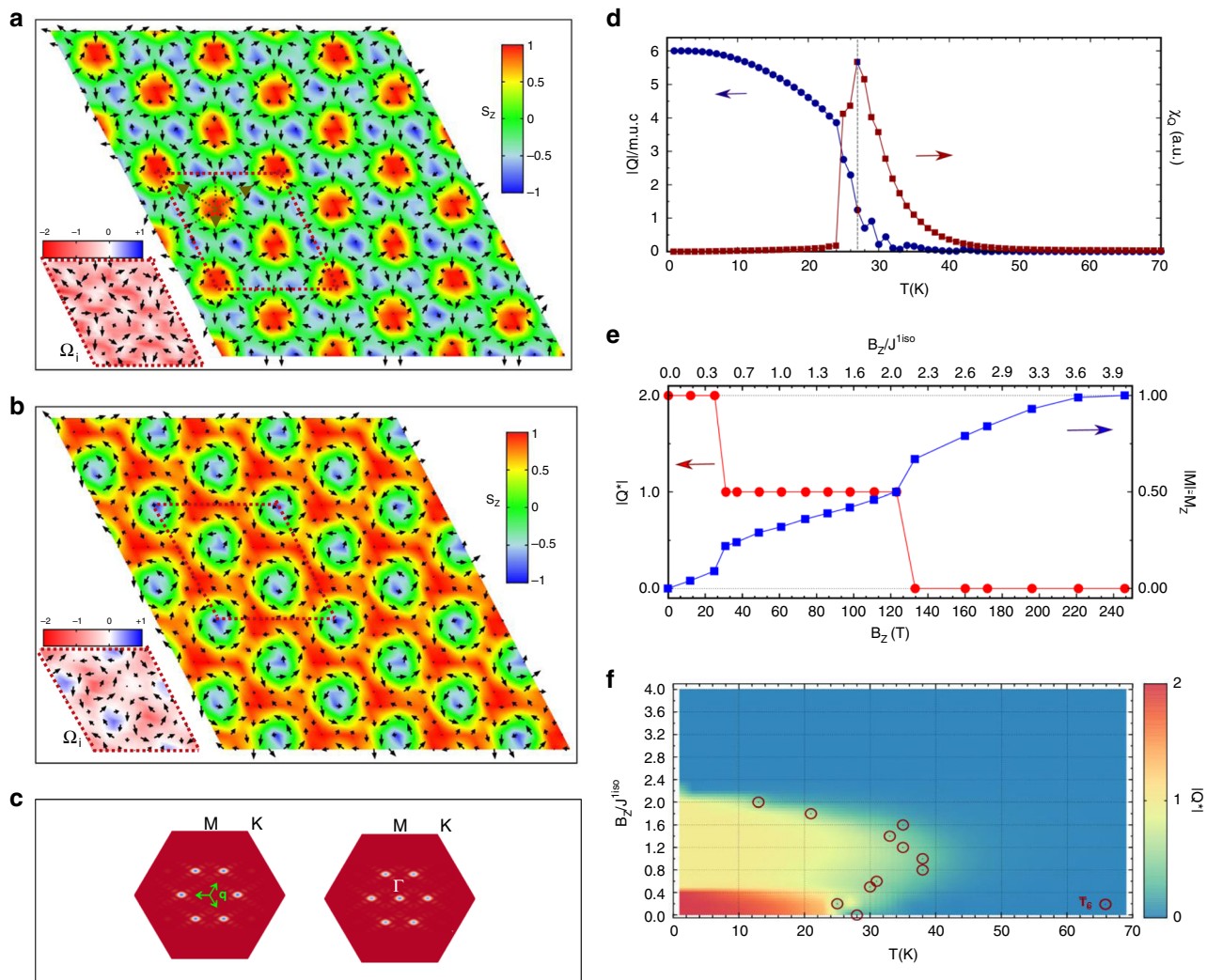

**Fig. 1 Spin structures and field-induced topological transition of the thermodinamically phases in monolayer NiI$_2$. a, b** Snapshots of spin configurations at $T = 1$ K from MC simulations on a 24 × 24 supercell obtained for $B_z/J^{1iso} = 0$ and $B_z/J^{1iso} \simeq 1.5$, displaying the antibiskyrmion and Bloch-skyrmion lattice, respectively. Black arrows represent in-plane components of spins, colormap indicates the out-of-plane ($s_z$) spin components. The magnetic unit cell is shown with dashed lines; dashed arrows in **a** are guidelines for the eyes to visualize the spins orientation and directions defining the antibiskyrmion. Insets show correspondent topological charge densities $\Omega_i$ in the selected magnetic unit cell. **c** The spin structure factor $S(\mathbf{q})$, corresponding to a triple-$\mathbf{q}$ state with $\mathbf{q}_1 = (\delta, \delta)$, $\mathbf{q}_2 = (\delta, -2\delta)$, and $\mathbf{q}_3 = (-2\delta, \delta)$ in the hexagonal setting, as highlighted by green arrows, with $\delta \simeq 0.125$. $S(\mathbf{q})$ on the left refers to the A2Sk state; $S(\mathbf{q})$ on the right refers to the Sk state with the additional peak at $\Gamma$ reflecting the ferromagnetic component induced by the applied magnetic field. **d** Topological charge $|Q|$ (closed circles) per magnetic unit cell and corresponding topological susceptibility $\chi_Q$ (closed squares) as a function of temperature for $B_z/J^{1iso} = 0$, pointing to a transition temperature $T_c \simeq 28$ K within the used DFT approximations. **e** Evolution of $|Q|$, normalized to the number of skyrmionic objects in the magnetic unit cell and of magnetization $M_z$ as a function of the magnetic field $B_z$ at $T = 1$ K. **f** Phase diagram in the temperature-field plane. Colormap indicates the topological charge densities $|Q|$ as defined in **e**. Critical temperatures ($T_c$) for topological phase transitions are also reported in circles.

analogy with the mechanism devised by Moriya for canted magnetism arising from different single-ion anisotropy axes[47].

In Table 1 we report such contributions to the exchange coupling for NiI$_2$ monolayer as obtained by DFT-calculations. Further insights on the properties of the exchange coupling tensor are discussed in Section SI of the Supplementary. Moreover, in order to highlight chemical trends, we also report results for the isostructural NiCl$_2$ and NiBr$_2$ monolayer systems.

This allows, on the one hand, to analyze the effects of different SOC strengths (going from the weak contribution expected in Cl to the strongest one in I) and, on the other hand, to explore the range of the interactions (going from the localized 3$p$ states in Cl to broader 5$p$ in I).

As evident from Table 1, the FM nearest-neighbor exchange interaction ($J^{1iso}$) becomes larger when moving from Cl to I. Interestingly, the resulting AFM third-nearest neighbor interaction ($J^{3iso}$) is also strongly affected, as a consequence of the broader ligand $p$ states mediating the superexchange[48]: the $J^{3iso}/J^{1iso}$ ratio is $\simeq -0.33$, $-0.49$, $-0.83$ for NiCl$_2$, NiBr$_2$, and NiI$_2$, respectively, revealing thus increasing magnetic frustration as a function of the ligand. A similar trend is observed for the exchange anisotropy, that also increases as the ligand SOC gets stronger along the series. Indeed, magnetism of NiCl$_2$ results well described by the isotropic Heisenberg model, as opposed to NiBr$_2$ and NiI$_2$. The strongest effect is predicted for NiI$_2$, where the two-site anisotropy associated to the nearest-neighbor exchange interaction is one order of magnitude larger than in NiBr$_2$. In

**Table 1 Exchange coupling parameters for NiCl$_2$, NiBr$_2$ and NiI$_2$ monolayers.**

|  | $J^{1iso}$ | $J^{2iso}$ | $J^{3iso}$ |
|---|---|---|---|
| NiCl$_2$ | −5.1 | −0.1 | +1.7 |
| NiBr$_2$ | −5.9 | −0.1 | +2.9 |
| NiI$_2$ | −7.0 | −0.3 | +5.8 |

| | $J^{two\text{-}site\ aniso}$ | | | | | |
|---|---|---|---|---|---|---|
| | $J_{xx}$ | $J_{yy}$ | $J_{zz}$ | $J_{yz}$ | $J_{xz}$ | $J_{xy}$ |
| NiCl$_2$ | 0.0 | 0.0 | 0.0 | 0.0 | 0.0 | 0.0 |
| NiBr$_2$ | −0.1 | +0.1 | 0.0 | −0.1 | 0.0 | 0.0 |
| NiI$_2$ | −1.0 | +1.4 | −0.3 | −1.4 | 0.0 | 0.0 |

| | $\lambda_\alpha$ | $\lambda_\beta$ | $\lambda_\gamma$ |
|---|---|---|---|
| NiCl$_2$ | −5.1 | −5.1 | −5.1 |
| NiBr$_2$ | −6.0 | −6.0 | −5.7 |
| NiI$_2$ | −8.1 | −8.0 | −4.8 |

(upper table) Isotropic first ($J^{1iso}$), second ($J^{2iso}$), and third ($J^{3iso}$) nearest neighbor interactions. (middle table) Matrix elements, in the cartesian $\{x, y, z\}$ reference, of the anisotropic symmetric exchange ($\mathbf{J}^S$) or two-site anisotropy ($\mathbf{J}^{two\text{-}site\ aniso}$) between first nearest-neighbor spins. Site indices (ij) are omitted. Values refer to the Ni$_0$ − Ni$_1$ pair, with the Ni-Ni bonding vector parallel to the cartesian axis $x$. (bottom table) Principal values (eigenvalues) of the full first nearest-neighbors exchange tensor, that is $\mathbf{J}^1 = J^{1iso}\mathbf{I} + \mathbf{J}^{two\text{-}site\ aniso}$. Exchange parameters are expressed in term of energy unit (meV).

particular, the $J_{yz}/J^{1iso}$ ratio, measuring the canting of the two-site anisotropy axes from the direction perpendicular to the monolayers, as argued below and in Section SI of the Supplementary, changes from 0.00 in NiCl$_2$ to 0.02 and 0.20 in NiBr$_2$ and NiI$_2$, respectively. This specific anisotropic contribution tends to vanish in the second-nearest and third-nearest neighbor exchange interactions. Moreover, second-nearest-neighbor and beyond third-nearest-neighbor interactions are at least one order of magnitude smaller than $J^{1iso}$ and $J^{3iso}$. The SIA (whose values are reported in Supplementary Table SI) is also negligible with respect to these two main interactions and the two-site anisotropy; the largest value of SIA was found for NiI$_2$, that is predicted to display easy-plane anisotropy with $(A_{zz} − A_{xx}) \simeq +0.6$ meV, the latter being not expected to play a relevant role in the stabilization of the triple-**q** state. In the following we will therefore focus on the crucial role played by the two-site anisotropy in driving the A2Sk lattice in NiI$_2$.

The exchange tensor discussed until now was expressed in the cartesian $\{x, y, z\}$ basis, where $x$ was chosen to be parallel to the Ni-Ni bonding vector (see Fig. 2a and Fig. S1 in Supplementary). In order to better understand and visualize the role of the two-site anisotropy, it is useful to express the interaction within the local principal-axes basis $\{\boldsymbol{v}_\alpha, \boldsymbol{v}_\beta, \boldsymbol{v}_\gamma\}$ which diagonalize the exchange tensor, with eigenvalues $(\lambda_\alpha, \lambda_\beta, \lambda_\gamma)$, as detailed in Section SI of Supplementary. Principal values of the exchange tensor for the three compounds are reported in Table 1. It is also noteworthy that, within this local basis, the exchange interaction could be further decomposed into an isotropic parameter $J'$ and a Kitaev term $K$, as in refs. [49,50] and Supplementary-Section SI, thus providing another estimate of the global anisotropy of the exchange interaction; the $|K/J'|$ ratio in fact evolves as $\simeq 0.00, 0.05, 0.40$ in NiCl$_2$, NiBr$_2$, and NiI$_2$, respectively.

In Fig. 2 we show the principal axes for the most anisotropic system, NiI$_2$: $\boldsymbol{v}_\alpha$ and $\boldsymbol{v}_\beta$ vectors lie in the Ni-I–Ni-I spin–ligand plaquette while $\boldsymbol{v}_\gamma$ is perpendicular to it (cfr Fig. 2b). Therefore, the six $\boldsymbol{v}_\alpha$ and $\boldsymbol{v}_\gamma$ vectors, being not parallel to any lattice vector, introduce a noncoplanar component in the interaction of the spins (Fig. 2c). This non-coplanarity is mathematically determined by the off-diagonal terms of the two-site anisotropy (see Section SI of Supplementary), arising from the SOC of the heavy I

ligand, combined with the noncoplanarity of the spin–ligand plaquettes on each triangular spin-spin plaquette displayed in Fig. 2c. Such noncoplanarity is the first main ingredient introduced by the two-site anisotropy and necessary to define the net scalar chirality of the spin texture. Its second main effect is also to fix the helicity, driving the in-plane orientation of the spins. The topology and chirality of the spin texture is thus well defined. In fact, when looking at the in-plane projection of the $\boldsymbol{v}_\alpha$ eigenvector reported in Fig. 2d, one can observe a direct relation to the anti-biskyrmion spin-pattern represented in Fig. 2e, f: spins on the nearest-neighbor Ni atoms surrounding the central spin orient in plane according to the in-plane components of the noncoplanar $\boldsymbol{v}_\alpha$ vector both in direction and sense (Fig. 2e, f); the accommodation of the anti-biskyrmions in the spin lattice, along with the conservation of a vanishing net magnetization in the system, determine then the direction and orientation of the second-neighbor spins, and, as a consequence, the pattern of the associated magnetic defects (i.e., the vortices) surrounding the A2Sk-core. In particular, the rotational sense of the spins, i.e., the chirality, is determined by the sign of the off-diagonal terms of the two-site anisotropy. We verified this dependence by artificially changing the sign of $J_{yz}$ term in the Ni$_0$–Ni$_1$ exchange tensor (that by symmetry also affects related terms of the other Ni–Ni coupling, as in Section SI-Supplementary) in the MC simulation; such operation would correspond to apply a reflection with respect to the $xy$ plane. The resulting topological lattice still displays vorticity $m = -2$, i.e., the A2Sk lattice, but with opposite chirality: indeed, the in-plane reflection leads to a change by $\pi$ of the helicity, not affecting the $z$-component of the magnetization and keeping the orientation of the spin at the core of each A2Sk (see Fig. S4 in Supplementary). Moreover, to further verify the unique relation between the two-site anisotropy and the topology of the spin structure, we also artificially swapped the exchange interaction of the Ni$_0$–Ni$_2$ and Ni$_0$–Ni$_3$ pairs obtaining now a lattice with opposite vorticity, $m = 2$, i.e., a biskyrmion (2Sk) lattice. Such spin state can be obtained from the A2Sk lattice by inverting the sign of the $x$-component of each spin, namely $(s_x, s_y, s_z) \rightarrow (−s_x, s_y, s_z)$, thus leading to an opposite sign of the scalar spin chirality and a consequent change of the spin-structure topology (see Fig. S5 in Supplementary). In this case, an applied magnetic field would stabilize an antiskyrmion lattice ($m = -1$), as shown in Fig. S5e, f.

**Competing magnetic frustrations.** It is important to stress that magnetic frustration is a necessary prerequisite for the stabilization of the skyrmionic triple-**q** state, further determining its periodicity and, hence, the size of individual topological objects (as shown in Fig. S7), whose character is then dictated by the frustrated two-site anisotropy. Indeed, the latter competes with the isotropic exchange that, in the absence of magnetic frustration, would favor collinear spin configurations. This is, for instance, the case of the prototypical 2D ferromagnet CrI$_3$, which displays an easy-axis FM ground-state despite the strong exchange anisotropy[49,51]. Indeed, Xu and coauthors[49] reported a strong anisotropic symmetric exchange for monolayer CrI$_3$, akin to what we obtained for monolayer NiI$_2$ and consistently with the similar metal-halide arrangement mediating the exchange inter-action. Nevertheless, in CrI$_3$ the magnetic frustration due to third-neighbor interaction is rather weak[52], because of the honeycomb lattice adopted by the Cr cations, preventing the stabilization of a noncoplanar spin structure and favouring, instead, an easy-axis FM state. On the other hand, the presence of a strong two-site anisotropy is necessary to obtain the topological triple-**q** state, when magnetic frustration is already strong enough to stabilize a noncollinear spin structure. In fact, in monolayer

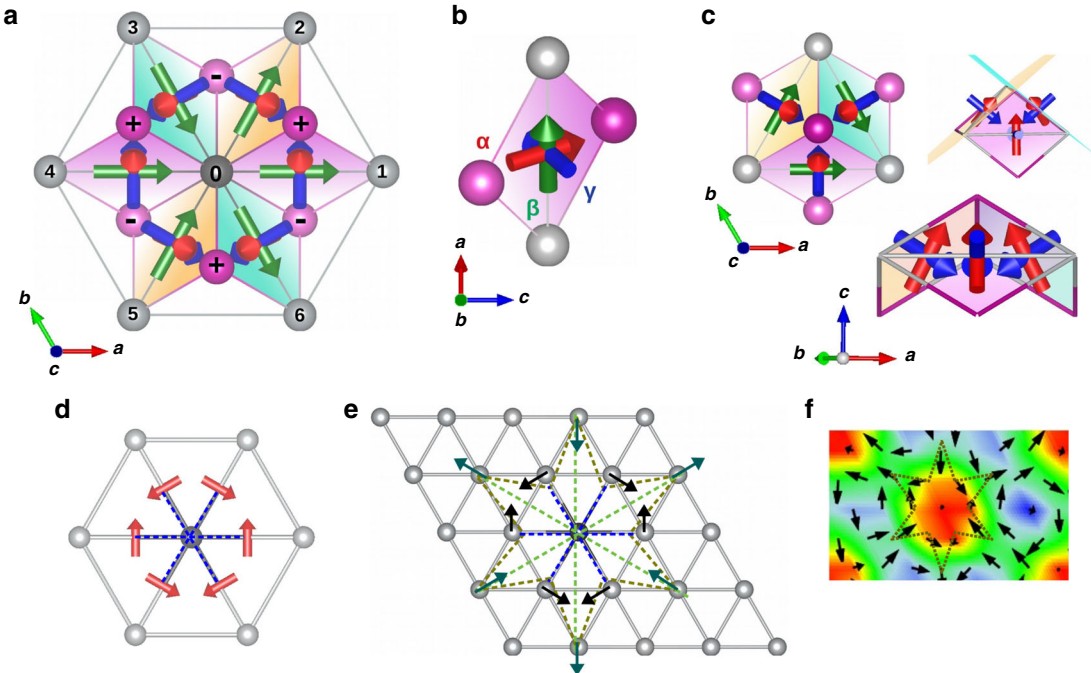

**Fig. 2 Schematic visualizations of the magnetic eigenvectors and their relation with the antibiskyrmion spin structure. a** Top view of $NiI_2$: the six nearest-neighbors Ni surrounding the central Ni atom are labeled with numbers. Plus and minus indicate iodine atoms above and below nickels plane, respectively. The $\{\nu_\alpha, \nu_\beta, \nu_\gamma\}$ eigenvector basis set is represented on each Ni–Ni pair. **b** Lateral view of the Ni–I–Ni–I plaquette and relative orientation of the eigenvectors: $\nu_\alpha$ ($\alpha$-red), $\nu_\beta$ ($\beta$-green), and $\nu_\gamma$ ($\gamma$-blue). **c** Top and lateral view of the local eigenvectors on the triangular Ni-net to help visualization of the noncoplanarity and noncollinearity in the exchange-tensor principal axes. **d** In-plane components of the $\nu_\alpha$ eigenvector for each magnetic Ni–Ni pair. **e** Sketches of the anti-biskyrmion spin structure: spins on the magnetic sites of the nearest-neighbors (black arrows) of the central Ni orient according to the in-plane projection of the noncoplanar principal axes (**d**); spins on the second-nearest neighbor magnetic sites (green arrows) orient following the direction fixed by the interaction accommodating the A2Sk in a spin lattice. Dashed gold lines are guidelines for the eyes delimiting the topological spin pattern and dashed blue (green) lines mark the direction of the six first (second) nearest neighbor Ni, as in Fig. S2c, d. **f** Zoom on the antibiskyrmion lattice as obtained from the MC simulations: spins orient as drawn in **e**.

$NiBr_2$, for which we found a much weaker contribution of the two-site anisotropy to the exchange interaction, the ground-state is a single-**q** helimagnetic state with no net scalar chirality (Fig. 3a). Its anisotropy still introduces frustration in the spins direction, but it is energetically not strong enough to stabilize the skyrmionic pattern; thus, a complex noncoplanar helimagnetic texture takes place, as detailed in Fig. 3a, c. Furthermore, neither the single-ion nor the two-site anisotropies are strong enough to support a field-driven topological transition to a skyrmion lattice; rather, a single-**q** conical cycloid state, shown in Fig. 3b, d, develops under applied field before a purely FM state is stabilized at $B_z/J^{1iso} \gtrsim 0.9$.

Further insight on the role of competing interactions is provided by comparing the total internal energy of $NiI_2$ evaluated on various aforementioned spin structures including first-neighbor and third-neighbor interaction parameters listed in Table 1 and the easy-plane single-ion anisotropy. As shown in Fig. 4a, the A2Sk lattice shows the lowest total energy among the possible topological triple-**q** states, namely the anti(bi)skyrmion and (bi)skyrmion lattices. As shown in Fig. 4b, the $|Q| = 1$ configurations (skyrmion, Sk, and antiskyrmion, ASk) are destabilized with respect to the $|Q| = 2$ ones (biskyrmion, 2Sk, and antibiskyrmion, A2Sk) mostly by the AFM third nearest neighbor interaction $J^{3iso}$, whose associated energy cost overcomes the energy gain due to FM nearest-neighbor exchange $J^{1iso}$. On the other hand, the two-site anisotropy $J^{two-site\ aniso}$ is responsible for the stabilization of the A2Sk lattice over the 2Sk one, since both would be energetically degenerate in the presence of purely isotropic (and competing)

exchange interactions. Furthermore, both the A2Sk and the Sk lattices can be realized with different chiralities of the spin configurations, related by an in-plane reflection, whose degeneracy is lifted by the two-site anisotropy, as shown in Fig. 4b. Due to their specific multichiral nature[1,20,53], no such chiral-degeneracy lifting is observed for the ASk and 2Sk lattices. Our results thus demonstrate that the two-site anisotropy may behave as an emergent chiral interaction for this class of centrosymmetric systems, determining a unique topology and chirality of the spin structure.

We also considered the total energies of possible single-**q** spin configurations, namely a coplanar cycloidal helix (with spins rotating in the $xy$ plane, consistently with the SIA easy-plane anisotropy) and a noncoplanar helix maximizing the two-site anisotropy energy gain, analogous to the complex helimagnetic state predicted for $NiBr_2$. For $NiI_2$, such spin configurations display an overall energy gain associated to isotropic exchanges with respect to the triple-**q** states, at the expense of an energy cost associated with the two-site anisotropy, as shown in Fig. 4b. Therefore, the competition bewteen isotropic and anisotropic symmetric exchanges determines which spin state is eventually stabilized. Interestingly, the noncoplanar helix is found to be almost degenerate with the A2Sk lattice, implying that the thermodynamic stability of the latter in $NiI_2$ has to be ascribed to thermal fluctuations and entropic contributions to the free energy accounted for in our MC simulations. Finally, the chiral nature of the two-site anisotropy emerges also for single-**q** states, lifting the degeneracy between the two chiral partners of the noncoplanar helix.

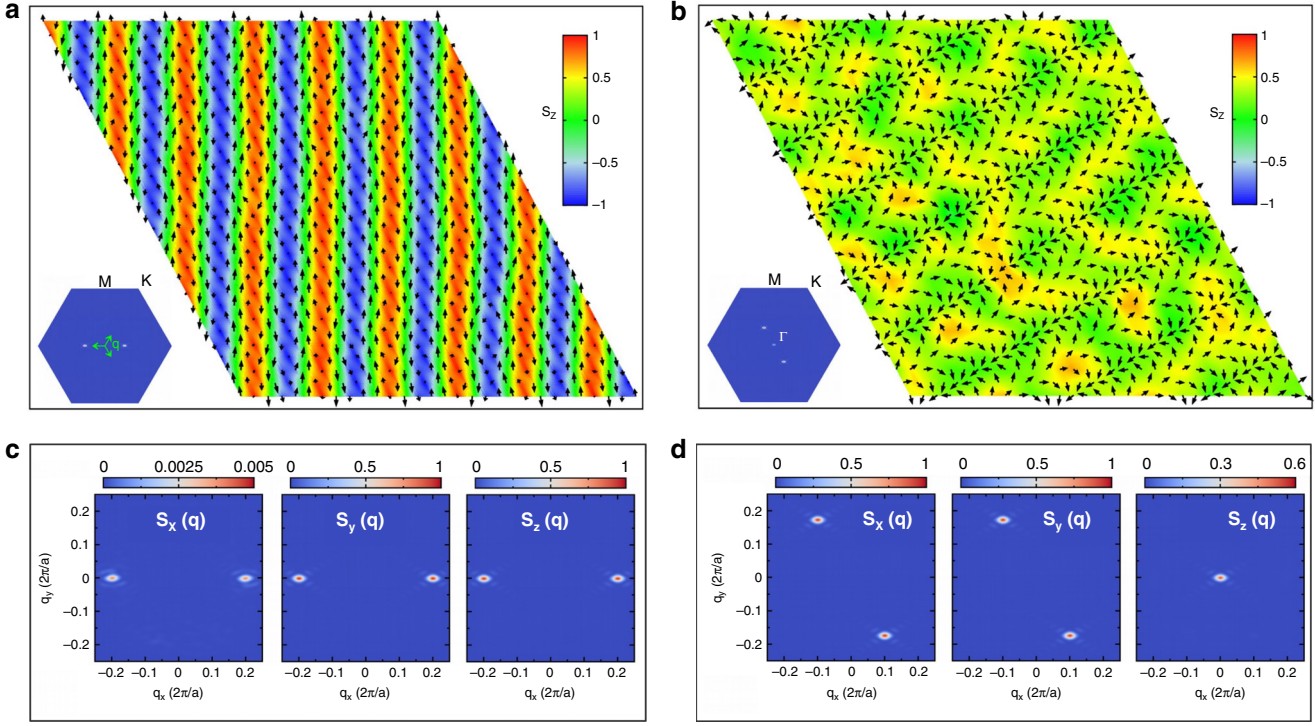

**Fig. 3 Magnetic structures for NiBr$_2$ monolayer. a, b** Snapshots of spin configurations at $T = 1$ K from MC simulations on a 30 × 30 supercell obtained for $B_z/J^{liso} = 0$ and $\simeq 0.3$, respectively. Insets show the spin structure factor $S(\mathbf{q})$, which corresponds to a single-$\mathbf{q}$ state along $\mathbf{q}_3 = (-2\delta, \delta)$ in **a** and $\mathbf{q}_2 = (\delta, -2\delta)$ in **b**, with $\delta \simeq 0.10$. **c, d** Decomposition of the spin structure factor in the cartesian components $S_x(\mathbf{q})$, $S_y(\mathbf{q})$ and $S_z(\mathbf{q})$. The spin configuration displayed in **a** is composed of a spin-spiral, with spins rotating in a plane perpendicular to the propagation vector $\mathbf{q}_3$, and of a spin-cycloidal component, with spins rotating in a plane containing the propagation vector, as highlighted by the small peak in $S_x(\mathbf{q})$. Such noncoplanar helimagnetic single-$\mathbf{q}$ state arises from the small anisotropic symmetric exchange reported in Table 1. Under an applied magnetic field, a conical helix state is stabilized (**b**), where a ferromagnetic component parallel to the field is superimposed to a cycloidal state, as evidenced by in-plane and out-of-plane components of the spin structure factor shown in **d**.

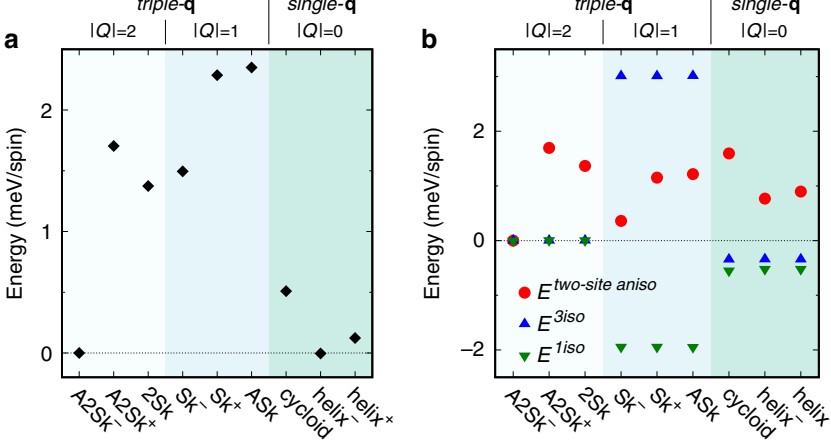

**Fig. 4 Total internal energies of selected spin configurations for NiI$_2$. a** Total internal energies of triple-$\mathbf{q}$ states with topological charge $|Q| = 2$ and $|Q| = 1$ and of selected single-$\mathbf{q}$ states. All energies, given with respect to the lowest-energy A2Sk state in units of meV/spin, have been calculated for NiI$_2$ using a 24 × 24 supercell with periodic boundary conditions. The superscript ± denotes the chiral partners of the given spin texture, related by a reflection with respect to the $xy$ plane (and consistent with the sign of the off-diagonal terms stabilizing the given spin configuration, as discussed in the text). The single-$\mathbf{q}$ spin configurations have been generated by MC simulations artificially tuning the two-site anisotropy term. **b** Contributions to the total internal energy $E^{liso}$, $E^{3iso}$, and $E^{two-site\ aniso}$ due to isotropic and anisotropic exchange interactions parametrized by $J^{liso}$, $J^{3iso}$ and $\mathbf{J}^{two-site\ aniso}$, respectively, given with respect to the lowest-energy A2Sk state in units of meV/spin.

## Discussion

In this work, we have identified the two-site anisotropy, arising from the short-range symmetric exchange interaction, as the driving mechanism for the stabilization of a spontan-eous topological spin-structure. Specifically, we predict a sponta-neous antibiskyrmion lattice below $T_c \simeq 30$ K along with a field-induced topological transition in NiI$_2$ monolayer, representative of the class of 2D magnetic semiconductors.

We found that metastable multi-$\mathbf{q}$ skyrmionic states, that may occur in frustrated magnets, can stabilize as ground-state with well-defined topology and chirality in the presence of competing two-site anisotropies characterized by noncoplanar principal axes. Such kind of additional frustration in the relative orientation of spins acts as an emergent chiral interaction, fixing the topology and the chirality of the localized spin textures and of the resulting skyrmion lattice, whereas its size and periodicity are mostly determined by competing isotropic exchanges. Interestingly, our findings are not limited to centrosymmetric systems, as the two-site anisotropy can be found in noncentrosymmetric magnets as well; its competition with the Dzyaloshinskii–Moriya interaction could thus also reveal interesting phenomena in many other systems, including the recently proposed Janus $Cr(I,Br)_3$ monolayer[50].

In conclusion, the proposed mechanism, which we have predicted here in a Ni-halide monolayer, enlarges both the kind of magnetic interactions able to drive the stabilization of topological spin structures, and the class of materials able to host spontaneous skyrmionic lattices with definite chirality, including also magnetic semiconductors with short-range anisotropic interactions.

## Methods

**First-principles calculations**. Magnetic parameters of the interacting spin Hamiltonian (1) were calculated by performing first-principles simulations within the density functional theory (DFT), using the projector-augmented wave method as implemented in the VASP code[54,–56]. The following orbitals were considered as valence states: Ni 3$p$, 4$s$ and 3$d$, Cl 3$s$ and 3$p$, Br 4$s$ and 4$p$, and I 5$s$ and 5$p$. The Perdew–Burke–Erzenhof (PBE) functional[57] within the generalized gradient approximation (GGA) was employed to describe the exchange-correlation (xc) potential; the plane wave cutoff energy was set to 600 eV for $NiCl_2$ and $NiBr_2$, and 500 eV for $NiI_2$, which is more then 130% larger than the highest default value among the involved elements. The $U$ correction[58] on the localized 3$d$ orbitals of Ni atoms was also included. Exchange energies reported in this article, and used to run the MC simulations, were calculated by employing $U = 1.8$ eV and $J = 0.8$ eV within the Liechtenstein approach[59]. We also adopted the Dudarev approach[60] to test the results solidity against different effective $U$ values; we choose $U$ equal to 1, 2, and 3 eV and a fixed $J$ equal to 0 eV. Results remain qualitatively the same: the $J^{3iso}/J^{1iso}$, and $J_{yz}/J^{1iso}$ ratios (i.e., the quantities with relevant physical meaning, even more than the absolute values of the interactions) are almost unaffected, as shown in Table SII in Supplementary.

We calculated the magnetic parameters reported in this paper via the four-state energy mapping method, which is explained in detail in the refs. [43,49,50], performing noncollinear DFT calculations plus spin–orbit coupling (SOC) and constraints on magnetic moments direction. It is based on the use of large supercells, also allowing to exclude the coupling with unwanted distant neighbors. By means of this method we can obtain all the elements of the exchange tensor for a chosen magnetic pair, thus gaining direct access to the symmetric anisotropic exchange part (the two-site anisotropy) and the antisymmetric anisotropic part (the DM interaction) of the full exchange. In particular, we performed direct calculations on the magnetic Ni–Ni pair parallel to the $x$ direction, here denoted $Ni_0$–$Ni_1$ (Fig. 2a). The interaction between the five other nearest-neighbor pairs can be evaluated via the three-fold rotational symmetry, as commented in Supplementary-Section SI. In all our systems, the tensor turned out to be symmetric or, equivalently, excluding any anti-symmetric (DM-like) contribution.

We performed calculations of the SIA, first-neighbors and second-neighbors interaction using a $5 \times 4 \times 1$ supercell; while a $6 \times 3 \times 1$ supercell for the estimate of the third-neighbors interaction. Such cells should exclude a significant influence from next neighbors. We built supercells from the periodic repetition of the $NiX_2$ monolayer unit cell, with lattice parameters ($a_0$) and ionic positions optimized by performing standard collinear DFT calculations with a ferromagnetic spin ordering. The obtained lattice parameters are: 3.49 Å and 3.69 Å for $NiCl_2$ and $NiBr_2$, respectively[61], which are in agreement (within 0.3% uncertainty) with the values known for the bulk compounds, and 3.96 Å for $NiI_2$, which is 1.5% larger than the 3.89 Å bulk value[30]. We thus checked the stability of the $NiI_2$ magnetic parameters by extracting them from a cell with lattice constant fixed to the experimental value, not obtaining significant changes (as reported in Table SIII in Supplementary). In all cases, the length of the out-of-plane axis, perpendicular to the monolayer plane, was fixed to 20.8 Å, which provides a distance of more than 17.5 Å with respect to the periodic repetition of the layer along this direction. The sampling of the Brillouin zone for the monolayer unit cell relied on a $18 \times 18 \times 1$ $k$-points mesh; meshes for the supercells have been chosen according to the latter.

**MC simulations**. MC calculations were performed using a standard Metropolis algorithm on $L \times L$ triangular supercells with periodic boundary conditions.

At each simulated temperature, we used $10^5$ MC steps for thermalization and $5 \times 10^5$ MC steps for statistical averaging. Average total energy, magnetization and specific heat have been calculated. The lateral size of the simulation supercells has been chosen as $L = nL_{m.u.c.}$, where $n$ is an integer and $L_{m.u.c.}$ is the lateral size of the magnetic unit cell needed to accomodate the lowest-energy noncollinear helimagnetic spin configurations. Accordingly, we estimated $L_{m.u.c.}$ as $1/q$, where $q$ is the length of the propagation vector $\mathbf{q}$ minimizing the exchange interaction in momentum space $J(\mathbf{q})$. For the isotropic model and neglecting second nearest-neighbor interactions, the propagation vector is given by $q = 2\cos^{-1}[(1 + \sqrt{1 - 2J^{1iso}/J^{3iso}})/4]$[14,21], resulting in $L_{m.u.c.} \simeq 8$ and $\simeq 10$ for $NiI_2$ and $NiBr_2$, respectively. Results are shown for calculations performed on supercells with lateral size $L = 3L_{m.u.c.}$, but we verified the accuracy of our choice by performing benchmark calculations on commensurate and incommensurate cells with $L$ ranging from 8 to 64. Further insight on the magnetic configurations was obtained by evaluating the spin structure factor:

$$S(\mathbf{q}) = \frac{1}{N} \sum_{\alpha=x,y,z} \left\langle \left| \sum_i s_{i,\alpha} e^{-i\mathbf{q}\cdot\mathbf{r}_i} \right|^2 \right\rangle, \qquad (2)$$

where $\mathbf{r}_i$ denotes the position of spin $\mathbf{s}_i$ and $N = L^2$ is the total number of spins in the supercell used for MC simulations. The braket notation is used to denote the statistical average over the MC configurations. The spin structure factor provides direct information on the direction and size of the propagation vectors, which are found to agree with the analytical estimate provided above.

In order to assess the topological nature of the multiple-$q$ phase, we evaluate the topological charge (skyrmion number) of the lattice spin field of each supercell as $\langle Q \rangle = \langle \Sigma_i \Omega_i \rangle$, where $\Omega_i$ is calculated for each triangular plaquette as[62]:

$$\tan\left(\frac{1}{2}\Omega_i\right) = \frac{\mathbf{s}_1 \cdot \mathbf{s}_2 \times \mathbf{s}_3}{1 + \mathbf{s}_1 \cdot \mathbf{s}_2 + \mathbf{s}_1 \cdot \mathbf{s}_3 + \mathbf{s}_2 \cdot \mathbf{s}_3}. \qquad (3)$$

The corresponding topological susceptibility has been evaluated as:

$$\chi_Q = \frac{\langle Q^2 \rangle - \langle Q \rangle^2}{k_B T}. \qquad (4)$$

## Data availability

Main results are reported in this article and related Supplementary Material. All other data that support the findings discussed in this study are available from the corresponding author upon reasonable request.

## Code availability

First-principles calculations were performed using the licensed VASP code[56]. Monte Carlo calculations were performed using a code implementing the standard Metropolis algorithm, that is available from the authors upon reasonable request.

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

## Acknowledgements

This work was supported by the Nanoscience Foundries and Fine Analysis (NFFA-MIUR Italy) project. P.B. and S.P. acknowledge financial support from the Italian Ministry for Research and Education through PRIN-2017 projects "Tuning and understanding Quantum phases in 2D materials—Quantum 2D" (IT-MIUR Grant No. 2017Z8TS5B) and "TWEET: Towards Ferroelectricity in two dimensions" (IT-MIUR Grant No. 2017YCTB59), respectively. Calculations were performed on the high-performance computing (HPC) systems operated by CINECA, supported by the ISCRA C (IsC66-I-2DFM, IsC72-2DFmF) and ISCRA B (IsB17-COMRED, grant HP10BSZ6LY) projects. We thank Krisztian Palotás, Bertrand Dupé, Mario Cuoco, Changsong Xu, Hongjun Xiang, Laurent Bellaiche, Hrishit Banerjee, Roser Valentí, Sang Wook Cheong, Sergey Artyukhin, and Stefan Blügel for helpful and illuminating discussions.

## Author contributions

D.A. performed first-principles calculations. P.B. and D.A. performed Monte Carlo simulations and analyzed the results. S.P. supervised the project. All the authors discussed the results and contributed to the manuscript writing.

## Competing interests

The authors declare no competing interests.
