## [Peer Review File · Nature Communications]

Reviewers' Comments:

Reviewer #1:

Remarks to the Author:

The work reported by Amoroso et al. entitled “*Spontaneous skyrmionic lattice from anisotropic symmetric exchange in a Ni-halide monolayer*”, manuscript # NCOMMS-20-18046, studies a spontaneous anti-biskyrmion lattice formed in a semiconducting metal halide (monolayer NiI₂) with centrosymmetric phase. With first-principle calculation and Monte Carlo simulation, they showed that the spin textures are because of emergent chiral interactions arising from a combination of anisotropic short-range symmetric exchange and magnetic frustration. Indeed, there have been recent reports on skyrmions in centrosymmetric systems, in addition to their formation in noncentrosymmetric ones with chiral magnets arising from the DM interactions.

The authors, in this study, have reported the occurrence of anti-biskyrmions in the monolayer of NiI₂, a 2D magnet. They report spontaneous formation of the given state below $T_c \sim 30\text{K}$. A magnetic anti-biskyrmion (A2sk) has six vortices in the boundary with S_z component in the boundary opposite to the S_z component in the center (which is well explained in Fig 1.a). Application of a perpendicular magnetic field (finite range) leads to creation of conventional skyrmionic lattice state and finally a FM (ferromagnetic state) at high enough magnetic field.

The authors have performed DFT calculations to find isotropic exchange interactions between first, second and third neighbors in the crystal lattice. The authors have also calculated anisotropic symmetric exchange/two site anisotropy. The strong AFM third neighbor interaction competes against the 1st neighbor interaction and hence leads to magnetic frustration. Moreover, the ratio between the isotropic exchange and the 2-site anisotropic exchange leads to the canting of the magnetic spins. Hence, magnetic frustration is the necessary ingredient for the A2sk state.

The work proposed in the present manuscript, is novel and timely, especially the 2D-magnets are being the newest member in 2D family, drawing significant scientific attention. However, I have the following comments upon addressing which the work could be suitable for publication in Nature Communications. Please see the following comments.

- The authors have found two-site anisotropic exchange from the isotropic exchange interaction to drive NiI₂ to spontaneously form topological spin states. The anti-biskyrmions are formed as a combination of anisotropic short-range symmetric exchange and magnetic frustration. Have the authors checked the presence and role of other kinds of interactions such as dipolar interactions etc.?
- A favorable condition for the considerations of these bi-skyrmions over individual skyrmions is the energy considerations, i.e., by comparing the total energy of possible spin textures. The authors did not provide a clear energy consideration to show the validity of the anti-biskyrmions in the main manuscript. It is recommended to provide an in-depth analysis of total energy considerations in various regimes discussed in their study.
- After citing ref. 4-10 in the first paragraph of introduction section in the manuscript but just before discussing the possibility of skyrmions in centrosymmetric systems, the authors are advised to cite a recent work on skyrmions in a 2D magnet, i.e., CrI₃ (Behera et al., APL 114, 232402 (2019)).
- Many of the notations, when first time used in the manuscript, should be briefly stated/defined with further reference to be made to the supplementary. In this sense, the article needs major update. This is very much needed for nonexperts in the field to broadly understand this important topic.
- $J_{1\text{iso}}$ and $J_{3\text{iso}}$ has been used in various instances throughout the manuscript but has not been clearly defined. More clarity is needed.

- Citation #21 is wrongly placed after the end of sentence (page 3).
- In the methods sections where the authors detailed about the Monte Carlo Simulations, they mentioned L between 10 and 60 but later for lowest energy configuration they took L as 8 and 10 for NiI₂ and NiBr₂ respectively. This is confusing.
- At each MC simulated temperature, authors used 10⁵ MC steps for thermalization and 5x10⁵ MC steps for statistical averaging. Why did they consider the difference for thermalization and statistical averaging?

Reviewer #2:

None

Reviewer #3:

Remarks to the Author:

The authors introduce a new mechanism for stabilizing a skyrmion crystal in a Ni-halide monolayer. The emergent chiral interaction arises from a finite canting of the two-site anisotropy axes from the direction perpendicular to the monolayers. This mechanism has not been reported before and, as the authors demonstrate, it is relevant for NiI₂ and presumably for other related halides. Consequently, I believe that this contribution is timely and important to guide the experimental search of exotic skyrmion crystal phases in monolayers. That said, there are a few aspects of the presentation of the manuscript that should be improved before I can recommend it for publication:

1) The authors include a single-ion anisotropy term in the Hamiltonian given in Eq.(1) but they omit to indicate the value of the tensor A. I think that this is an important omission because the single-ion anisotropy is known to be easy-plane in the bulk version of these materials. It is also known that the easy-plane anisotropy does not favor the stabilization of field-induced skyrmion crystals. This is probably the main reason why the bulk versions of these materials do not exhibit field-induced skyrmion crystal phases that would require easy-axis anisotropy to exist. The authors should then clarify the following points:

a) What is the value of the tensor A for each of the three monolayers that are discussed in the manuscript?

b) What is the effect of A on the phase diagram of NiI₂ that they are reporting in Fig.1?

c) In case A turns out to be easy-axis for any of the three materials under consideration, they authors may want to emphasize this fact since that alone would be enough to stabilize a field-induced skyrmion crystal [see for instance Rep Prog Phys. 2016;79(8):0845040].

2) By Fourier transforming the exchange interactions provided in Table I for each of the three compounds, the authors can compute the magnitude and possible directions of the ordering wave vector Q that is expected for each material [minimum of J(q)]. I suggest to include this information in the manuscript. The wavelength associated with such ordering wave vector determines both the linear size of the skyrmions and the periodicity of the Skyrmion crystals that are proposed for NiI₂.

3) In addition to the symmetric exchange anisotropy induced by the tilting of the local anisotropy axis, the authors obtain a "compass like" anisotropy term, which is proportional to $J_{yy} - J_{xx}$ for the Ni-Ni bond parallel to the x-axis. The value of this compass-like anisotropy seems to be correlated with the value of J_{yz} , which is responsible for the non-coplanar ordering. Can the authors comment on the role of the compass-like anisotropy on the stabilization of the Skyrmion phases that they are reporting?

4) The fact that effective 4-spin interactions, which are rather large in itinerant magnets, stabilize triple-Q orderings has been reported in papers that are actually older than the ones that the authors are citing. See for instance PRL 101, 156402 (2008); J. Phys. Soc. Jpn. 79, 083711 (2010); Rep Prog Phys. 2016;79(8):0845040 and references therein.

Point-by-point response to the reviewers for
Spontaneous skyrmionic lattice from anisotropic symmetric exchange
in a Ni-halide monolayer

Danila Amoroso,¹ Paolo Barone,¹ and Silvia Picozzi¹

¹*National Research Council CNR-SPIN, c/o Università degli Studi “G. D’Annunzio”, I-66100 Chieti, Italy*

In the following we report duly the “*Remarks to the Author(s)*” addressed by Reviewer#1 and Reviewer#3 along with our point-by-point responses.

I. REVIEWER#1

The work reported by Amoroso et al. entitled “*Spontaneous skyrmionic lattice from anisotropic symmetric exchange in a Ni-halide monolayer*”, manuscript #NCOMMS-20-18046, studies a spontaneous anti-biskyrmion lattice formed in a semiconducting metal halide (monolayer NiI₂) with centrosymmetric phase. With first-principle calculation and Monte Carlo simulation, they showed that the spin textures are because of emergent chiral interactions arising from a combination of anisotropic short-range symmetric exchange and magnetic frustration. Indeed, there have been recent reports on skyrmions in centrosymmetric systems, in addition to their formation in noncentrosymmetric ones with chiral magnets arising from the DM interactions.

The authors, in this study, have reported the occurrence of anti-biskyrmions in the monolayer of NiI₂, a 2D magnet. They report spontaneous formation of the given state below $T_c \sim 30\text{K}$. A magnetic anti-biskyrmion (A2sk) has six vortices in the boundary with S_z component in the boundary opposite to the S_z component in the center (which is well explained in Fig 1.a). Application of a perpendicular magnetic field (finite range) leads to creation of conventional skyrmionic lattice state and finally a FM (ferromagnetic state) at high enough magnetic field.

The authors have performed DFT calculations to find isotropic exchange interactions between first, second and third neighbors in the crystal lattice. The authors have also calculated anisotropic symmetric exchange/two site anisotropy. The strong AFM third neighbor interaction competes against the 1st neighbor interaction and hence leads to magnetic frustration. Moreover, the ratio between the isotropic exchange and the 2-site anisotropic exchange leads to the canting of the

magnetic spins. Hence, magnetic frustration is the necessary ingredient for the A2sk state.

The work proposed in the present manuscript, is novel and timely, especially the 2D-magnets are being the newest member in 2D family, drawing significant scientific attention. However, I have the following comments upon addressing which the work could be suitable for publication in Nature Communications.

We thank the referee for appreciating the novelty and timeliness of our work, as well as for raising constructive comments which are addressed below.

Please see the following comments.

1) The authors have found two-site anisotropic exchange from the isotropic exchange interaction to drive NiI_2 to spontaneously form topological spin states. The anti-biskyrmions are formed as a combination of anisotropic short-range symmetric exchange and magnetic frustration. Have the authors checked the presence and role of other kinds of interactions such as dipolar interactions etc.?

The main aim of our study was to carefully investigate the role of the anisotropic symmetric exchange contribution to the spin-spin Hamiltonian and its effects on the resulting spin configuration. We remark that anisotropic exchange is often neglected and its importance has been, so far, overlooked; therefore, we have not extended our investigation to other kinds of magnetic interactions, such as long-ranged dipolar interactions (that are currently not included in DFT codes, see Pellegrini et al. PRB 101, 144401 (2020) for a recent implementation of dipole-dipole interactions

FIG. 1: Snapshots of spin configurations for NiI_2 from MC simulations on a $L \times L = 24 \times 24$ supercell. The magnetic unit cell (m.u.c.) is shown with dashed lines - $L_{muc} \simeq 8$ - and it comprises three topological objects.

in DFT formalism). The found topological spin-states have been obtained from MC simulations by including only the aboved-mentioned contributions along with the single-ion anisotropy; as a consequence, no mechanism can be at the origin of the spontaneous formation of the A2Sk spin-configuration other than those included, *i.e.* the interplay between the two-site anisotropy and magnetic frustration. Moreover, dipolar interactions are expected to be relevant at the long-range scale and to give rise to rather large skyrmionic or bound-skyrmions spin-states with size ≥ 100 nm [Nature Nanotech.8, 899 (2013); Sci. Rep. 9, 9521 (2019)]. In the present case, the obtained topological objects are atomic-sized skyrmionic structures of the order of few lattice constants, as shown in Fig 1 (currently added to the revised Supplementary Material): the A2Sk-diameter counts $\simeq 5$ spins.

A possible competition with additional interactions could in principle affect the properties and relative stability of the topological spin-states. Nevertheless, it would not affect the found, intrinsic, chiral nature of the anisotropic short-range symmetric exchange, which is the main message that we would like to convey with our manuscript.

2) A favorable condition for the considerations of these bi-skyrmions over individual skyrmions is the energy considerations, *i.e.*, by comparing the total energy of possible spin textures. The authors did not provide a clear energy consideration to show the validity of the anti-biskyrmions in the main manuscript. It is recommended to provide an in-depth analysis of total energy considerations in various regimes discussed in their study.

We thank the Referee for this comment about energy considerations, which allowed us to improve our manuscript. We followed his/her suggestion by comparing the total (internal) energies of different spin textures evaluated on the model (defined in Eq. 1 of the main text) and using the interaction parameters estimated for NiI_2 . Namely, we considered all the (bi)skyrmion and anti(bi)skyrmion lattices discussed in the main text, as well as two single-q helical states (an in-plane cycloid and a non-coplanar helical state analogous to that we obtained for NiBr_2). All spin configurations have been obtained by MC calculations with a proper tuning of the interaction parameters, and validated by applying appropriate transformations to the spin components (*e.g.*, chiral partners of each spin configurations have been also obtained via a mirror operation about the xy plane; biskyrmion and antiskyrmion lattices have also been obtained by switching the x component of each spin of the antibiskyrmion and skyrmion lattices, respectively). We also decomposed such total energies in the contributions due to isotropic (first and third nearest neighbour exchange) and anisotropic interactions. Our additional analysis corroborates the key

role of the two-site anisotropy in stabilizing the A2Sk lattice over other possible triple-q topological states. Accordingly, we extended the section titled "Competing magnetic frustrations" including the aforementioned analysis of the main energy contributions.

3) After citing ref. 4-10 in the first paragraph of introduction section in the manuscript but just before discussing the possibility of skyrmions in centrosymmetric systems, the authors are advised to cite a recent work on skyrmions in a 2D magnet, i.e., CrI₃ (Behera et al., APL 114,232402 (2019)).

We followed the referee recommendation and added a citation to the suggested paper; however, since that work is focused on skyrmions in CrI₃ as related to DM induced by application of external electric field, we thought it was more appropriate to cite it in the list of references concerning 2D magnets (actual [31-40] list) having "intriguing low-dimensional magnetic properties"

4) Many of the notations, when first time used in the manuscript, should be briefly stated/defined with further reference to be made to the supplementary. In this sense, the article needs major update. This is very much needed for nonexperts in the field to broadly understand this important topic.

5) J_{1iso} and J_{3iso} has been used in various instances throughout the manuscript but has not been clearly defined. More clarity is needed.

We thank the Referee for his/her comments 4-5) aimed to improve the comprehension of the adopted notation, hence leading to a better overall understanding of our manuscript. Following his/her suggestion i) we clarified some unclear notations (for instance, J_{1iso} and J_{3iso} have been changed into J^{1iso} and J^{3iso} referring to the isotropic contribution to the exchange \mathbf{J} from the first and third nearest-neighbours, respectively, consistent with the notation introduced in Eq. 1); ii) we introduced more citations to previous papers reporting the well-established formalism adopted here (current citations [43,45-47]); iii) finally, we added further reference to the supplementary, where more details are given.

6) Citation #21 is wrongly placed after the end of sentence (page 3).

We thank the referee for this remark. We corrected it accordingly.

7) In the methods sections where the authors detailed about the Monte Carlo Simulations, they mentioned L between 10 and 60 but later for lowest energy configuration they took L as 8 and 10

for NiI_2 and NiBr_2 respectively. This is confusing.

We thank the Referee for highlighting the unclear formulation of the commented sentences. Our intent was to mention that we runned some MC simulations on various L-sized magnetic supercell in order to check the stability of the results. On the other hand, $L = 8$ and 10 correspond to the size of the magnetic unit cell for NiI_2 and NiBr_2 , respectively, accomodating the non-collinear/non-coplanar period modulation of the spin configurations. In order to avoid further confusion, in the revised version we denote as $L_{m.u.c.}$ the lattice constant of the magnetic unit cell, and we define as $L = nL_{m.u.c.}$ the lateral size of simulation supercells commensurate with the magnetic periodicity.

We rephrased the related sentences, clarifying the relationship between the size of supercells used for MC calculations and the size of the magnetic unit cell.

8) At each MC simulated temperature, authors used 10^5 MC steps for thermalization and 5×10^5 MC steps for statistical averaging. Why did they consider the difference for thermalization and statistical averaging?

The number of MC sweeps for thermalization is related to the time required by the system to reach “equilibrium” at a given temperature, where “equilibrium” means that the average probability of the system being in a particular state is proportional to its Boltzmann weight. Clearly, the states visited by the system during the thermalization process cannot be used for the statistical average of the physical quantities of interest, since they are not representative of the system at equilibrium. In Fig. 2, left panel, the evolution of the internal energy and topological charge as a function of MC sweeps is shown for $T = 1$ K, starting from a configuration of randomly-oriented spins, showing that at least $\tau_{eq} \sim 5 \times 10^4$ MC sweeps are required to achieve equilibrium. On the other hand, the choice of the number of MC sweeps used for averages depends on the desired statistical accuracy, where the latter is related to the number of statistically independent measurements included in the simulation. Statistical independence is achieved when two measurements are separated by a number of MC sweeps that is at least twice the correlation time τ_{corr} , being τ_{corr} the typical time-scale on which the time-displaced auto-correlation function of a given observable drops off. In the right panel of Fig. 2 we show the auto-correlation function of the topological charge evaluated at $T = 31$ K, which yields a correlation time $\tau_{corr} \simeq 360$ MC sweeps; this implies that the chosen number of MC sweeps for averaging guarantees that at least 700 independent measurements are used when evaluating the averaged topological charge.

FIG. 2: (Left) Internal energy per spin and topological charge per magnetic unit cell as a function of MC steps on a 24×24 cell at $T = 1$ K and evolving from a configuration of randomly-oriented spins. (Right) Auto-correlation function of the topological charge evaluate on thermalized states for a 24×24 cell at $T = 31$ K; the correlation time τ_{corr} has been extracted by fitting $\chi(t)$ with the exponential function $e^{-t/\tau_{corr}}$, shown as a red line.

II. REVIEWER#3

The authors introduce a new mechanism for stabilizing a skyrmion crystal in a Ni-halide monolayer. The emergent chiral interaction arises from a finite canting of the two-site anisotropy axes from the direction perpendicular to the monolayers. This mechanism has not been reported before and, as the authors demonstrate, it is relevant for NiI₂ and presumably for other related halides. Consequently, I believe that this contribution is timely and important to guide the experimental search of exotic skyrmion crystal phases in monolayers. That said, there are a few aspects of the presentation of the manuscript that should be improved before I can recommend it for publication:

We acknowledge the overall positive report of the Referee, who recognizes the importance, originality and timeliness of our work. We addressed all the issues he/she has raised in detail below:

1) The authors include a single-ion anisotropy term in the Hamiltonian given in Eq.(1) but they omit to indicate the value of the tensor A. I think that this is an important omission because the single-ion anisotropy is known to be easy-plane in the bulk version of these materials. It is also known that the easy-plane anisotropy does not favor the stabilization of field-induced skyrmion crystals. This is probably the main reason why the bulk versions of these materials do

not exhibit field-induced skyrmion crystal phases that would require easy-axis anisotropy to exist.

The authors should then clarify the following points:

- a) What is the value of the tensor A for each of the three monolayers that are discussed in the manuscript?
- b) What is the effect of A on the phase diagram of NiI₂ that they are reporting in Fig.1?
- c) In case A turns out to be easy-axis for any of the three materials under consideration, they authors may want to emphasize this fact since that alone would be enough to stabilize a field-induced skyrmion crystal [see for instance Rep Prog Phys. 2016;79(8):0845040].

We thank Reviewer#3 for addressing the question about the role of single-ion anisotropy (SIA).

Indeed, we did not omit to report values in our work, as from the following sentence in pag.3 of the original version of the manuscript:

“the SIA (whose values are reported in Supplementary-Table SI) is also negligible with respect to the two main interactions. In the following we will therefore focus on the crucial role played by the two-site anisotropy in driving the A2Sk lattice in NiI₂” .

In Table SI we reported the coupling constant A_i of the SIA contribution $H_{SIA} = \sum_i A_i S_{iz}^2$. Nevertheless, we have improved the presentation of our results regarding the full SIA tensor in the Supplementary, showing that indeed NiI₂ displays an easy-plane anisotropy, and we have further emphasized that SIA has negligible effects on the physics discussed in the main text.

In the following we then answer in detail to the a) b) c) points addressed by the Referee.

- a) What is the value of the tensor A for each of the three monolayers that are discussed in the manuscript?

As reported in Table SI :

- NiCl₂-monolayer displays no single-ion anisotropy within our numerical accuracy;
- NiBr₂-monolayer displays close-to-vanishing easy-plane single-ion anisotropy with value of +0.01 meV within the used DFT+U approximation with (U,J)=(1.8,0.8)eV.
- NiI₂-monolayer displays easy-plane single-ion anisotropy with value of +0.58 meV within the (U,J)=(1.8,0.8)eV DFT+U approximation.

In particular, for NiI₂, we also verified that no further terms appear in the expansion of the single-ion anisotropy Hamiltonian [H.Xiang et al. *Dalton. Trans.*, **42**, 823 (2013)]: coefficients A_{xy} , A_{xz} , A_{yz} , and $(A_{yy} - A_{xx})$ are in fact, within our numerical accuracy, equal to zero; only the $(A_{zz} - A_{xx})$ is $\simeq +0.6$ meV, resulting thus into an easy-plane anisotropy.

b) What is the effect of A on the phase diagram of NiI₂ that they are reporting in Fig.1?

We found that the SIA does not affect the spontaneous stabilization of the anti-biskyrmionic lattice, being its contribution negligible with respect to the major J^1 and J^3 interactions and also with respect to the J_{yz} anisotropic term ($|SIA/J_{1iso}| \simeq 0.08$; $|SIA/J_{3iso}| \simeq 0.10$; $|SIA/J_{yz}| \simeq 0.4$). In fact, when performing MC simulations by *i*) putting the SIA equal to zero and *ii*) changing its sign, in both cases the topology of the A2Sk-lattice, as well as the one of the excited Sk-lattice, were verified to be unaffected .

In deriving the phase diagram of NiI₂ reported in Fig. 1 in the main text, we included SIA using the calculated value. Even though we agree with the Referee that single-ion anisotropy may in general affect the properties and relative stability of topological spin-states, we didn't investigate the dependence of the phase diagram on it by artificially tuning the SIA value. We believe that this analysis, albeit interesting and possibly deserving further studies, would be beyond the scope of the present work, that focuses instead on a previously unreported mechanism for stabilizing skyrmionic lattices based on two-site anisotropy. For the sake of completeness, we highlighted that in our systems the SIA plays a secondary role both in the revised main text and in Supplementary material.

c) In case A turns out to be easy-axis for any of the three materials under consideration, they authors may want to emphasize this fact since that alone would be enough to stabilize a field-induced skyrmion crystal [see for instance Rep Prog Phys. 2016;79(8):0845040].

The presence of an easy-axis anisotropy can indeed favor the formation of skyrmions upon application of external magnetic field even in frustrated centrosymmetric systems, as we mentioned in the introduction of the original version of the manuscript:

“These states, that are metastable in classical isotropic systems, can be stabilized by applied fields, easy-axis magnetic anisotropy or long-range dipole-dipole and/or Ruderman-Kittel-Kasuya-Yosida (RKKY) interactions and thermal or quantum fluctuations [11-13, 15, 53]”.

Nevertheless, we thank the Referee for pointing out the review paper Rep Prog Phys. 2016;79(8):0845040, which we cited in the revised version. We remark, however, that no easy-axis anisotropy was found in the systems discussed in the main text, and we hope we made this point clearer in the revised version. In this respect, we also hope that the Referee will better appreciate the importance of two-site anisotropy in the proposed mechanism based on competing magnetic frustrations.

2) By Fourier transforming the exchange interactions provided in Table I for each of the three compounds, the authors can compute the magnitude and possible directions of the ordering wave vector \mathbf{Q} that is expected for each material [minimum of $J(\mathbf{q})$]. I suggest to include this information in the manuscript. The wavelength associated with such ordering wave vector determines both the linear size of the skyrmions and the periodicity of the Skyrmion crystals that are proposed for NiI₂.

We thank Referee#3 for his/her suggestion. Indeed, in the Methods section of the original version of our manuscript we provided an estimate of the magnitude of the ordering wave vector considering the Fourier transform $J(\mathbf{q})$ of isotropic exchange interactions, whose minimization leads to a simple analytical expression $q = 2 \cos^{-1}[(1 + \sqrt{1 - 2J_{1iso}/J_{3iso}})/4]$ (as also reported in Rep. Prog. Phys. 2016;79(8):084504). Since no such simple analytical expression was found when considering the full tensorial form of the exchange interactions, we calculated the spin structure factor $S(\mathbf{q})$ (as detailed in the Methods section of the original manuscript), providing direct information on the size and direction of the propagation vectors. We found good agreement between the numerical and analytical q estimates, suggesting that the magnetic periodicity is mostly determined by the dominant (isotropic) exchanges. In particular, in Fig. 1 and Fig. 3 of the original manuscript, we showed and commented the direction and amplitude of the propagation vector \mathbf{q} , as extracted from $S(\mathbf{q})$ for the skyrmionic lattices in NiI₂ and the helimagnetic states in NiBr₂ respectively. Moreover, a relation between the periodicity (or magnetic lattice constant, $L_{m.u.c.}$) and the radius of each anti-biskyrmion for the A2Sk-lattice was briefly commented in the caption of Fig. S7 (old Fig. S6), along with the relationship between the strength of the magnetic frustration and the size of the topological objects.

However, following the referee's comment suggesting the importance of such information, we put more emphasis on the periodicity of the spin-configurations even in the main-text, alongside the already detailed captions of Figs. 1,3, and we added a new explanatory figure in the Supplementary (present Fig. S6) about the anti-biskyrmions size.

3) In addition to the symmetric exchange anisotropy induced by the tilting of the local anisotropy axis, the authors obtain a "compass like" anisotropy term, which is proportional to $J_{yy}-J_{xx}$ for the Ni-Ni bond parallel to the x-axis. The value of this compass-like anisotropy seems to be correlated with the value of J_{yz} , which is responsible for the non-coplanar ordering. Can the authors comment on the role of the compass-like anisotropy on the stabilization of the Skyrmion phases that they are reporting?

To be rigorous, the tilting of the local anisotropy axis is determined by the full tensor describing

the two-site anisotropy, which includes not only the off-diagonal terms but also the anisotropy in the diagonal ones, to which he/she refers to as “compass anisotropy”. In this respect, the Referee is right when he/she suggests that the “compass-like” and off-diagonal terms are correlated. However, even though the final tilting angle is determined by the full tensor, a non-zero value of the J_{yz} off-diagonal term discriminates if such canting occurs or not. This result emerges clearly from the analytic expression of eigenvectors ($\boldsymbol{\nu}$) reported in Eq. S3 of Supplementary, which we report here along with corresponding eigenvalues (λ)

$$\left\{ \begin{array}{l} \lambda_\alpha = \frac{1}{2} \left(J_{yy} + J_{zz} - \sqrt{4J_{yz}^2 + (J_{yy} - J_{zz})^2} \right) \\ \lambda_\beta = J_{xx} \\ \lambda_\gamma = \frac{1}{2} \left(J_{yy} + J_{zz} + \sqrt{4J_{yz}^2 + (J_{yy} - J_{zz})^2} \right) \end{array} \right. , \left\{ \begin{array}{l} \boldsymbol{\nu}_\alpha = \left(0, -\text{sgn}(J_{yz})\sqrt{\frac{1-\cos\theta}{2}}, \sqrt{\frac{1+\cos\theta}{2}} \right) \\ \boldsymbol{\nu}_\beta = (1, 0, 0) \\ \boldsymbol{\nu}_\gamma = \left(0, \sqrt{\frac{1+\cos\theta}{2}}, \text{sgn}(J_{yz})\sqrt{\frac{1-\cos\theta}{2}} \right) \end{array} \right. \quad (1)$$

$$\text{where } \cos\theta = \frac{J_{yy} - J_{zz}}{\sqrt{4J_{yz}^2 + (J_{yy} - J_{zz})^2}} .$$

When considering the two extreme cases, that is either $J_{yz} = 0$ or $(J_{yy} - J_{zz}) = 0$, we obtain:

$$1) J_{yz} = 0 \rightarrow \cos\theta = 1$$

$$\left\{ \begin{array}{l} \lambda_\alpha = J_{zz} \\ \lambda_\beta = J_{xx} \\ \lambda_\gamma = J_{yy} \end{array} \right. \quad \left\{ \begin{array}{l} \boldsymbol{\nu}_\alpha = (0, 0, 1) \\ \boldsymbol{\nu}_\beta = (1, 0, 0) \\ \boldsymbol{\nu}_\gamma = (0, 1, 0) \end{array} \right.$$

implying coplanar principal axes on the Ni triangular lattice and locally parallel to the x, y, z cartesian axes, i.e. spins interaction independent on the Ni-I-Ni-I plaquettes, as shown in the inset. In this case, there are no terms able to drive the formation of a non-coplanar spin-texture; still, it can result into a topologically-trivial non-collinear spin-configuration, driven by the magnetic frustration, as it results from MC simulations tests.

$$2) (J_{yy} - J_{zz}) = 0 \rightarrow \cos\theta = 0 ; J_{xx} = J_{yy} = J_{zz} = J$$

$$\left\{ \begin{array}{l} \lambda_\alpha = J - J_{yz} \\ \lambda_\beta = J \\ \lambda_\gamma = J + J_{yz} \end{array} \right. \quad \left\{ \begin{array}{l} \boldsymbol{\nu}_\alpha = \left(0, -\text{sgn}(J_{yz})\sqrt{\frac{1}{2}}, \sqrt{\frac{1}{2}} \right) \\ \boldsymbol{\nu}_\beta = (1, 0, 0) \\ \boldsymbol{\nu}_\gamma = \left(0, \sqrt{\frac{1}{2}}, \text{sgn}(J_{yz})\sqrt{\frac{1}{2}} \right) \end{array} \right.$$

which still introduces non-coplanarity of principal axis, as shown in the inset. Nevertheless, having reduced the strength of the global anisotropy in terms of energy, the formation of the non-coplanar spin-texture, i.e. the A2Sk, would likely depend on the competition between the isotropic term and the anisotropic J_{yz} . This is confirmed from MC simulation test: in this extreme case, an increased value of J_{yz} – keeping fixed all other interaction terms – would be required to obtain the spontaneous formation of the A2Sk-lattice.

Since we think the above analysis might be useful to improve the overall understanding, we added it in Section I of the Supplementary.

4) The fact that effective 4-spin interactions, which are rather large in itinerant magnets, stabilize triple-Q orderings has been reported in papers that are actually older than the ones that the authors are citing. See for instance PRL 101, 156402 (2008); J. Phys. Soc. Jpn. 79, 083711 (2010); Rep Prog Phys. 2016;79(8):0845040 and references therein.

We thank the Referee for highlighting these papers. Accordingly, we added the suggested citations to PRL 101, 156402 (2008) and J. Phys. Soc. Jpn. 79, 083711 (2010). Reference Rep Prog Phys. 2016;79(8):084504 has been already considered with respect to her/his comment 1c) .

Reviewers' Comments:

Reviewer #1:

Remarks to the Author:

The authors have successfully addressed my previous comments. I feel and recommend the paper to be accepted for publication in its current form in Nature Communications.

Reviewer #3:

Remarks to the Author:

The authors have properly addressed the list of comments that I included in my previous report. Consequently, I recommend the new version of their manuscript for publication in Nature Communications.